# Evaluation and Determination of a Suitable Passage Number of Codon Pair Deoptimized PRRSV-1 Vaccine Candidate in Pigs

**DOI:** 10.3390/v15051071

**Published:** 2023-04-27

**Authors:** Min-A Lee, Su-Hwa You, Usharani Jayaramaiah, Eun-Gyeong Shin, Seung-Min Song, Lanjeong Ju, Seok-Jin Kang, Sun-Hee Cho, Bang-Hun Hyun, Hyang-Sim Lee

**Affiliations:** 1Viral Disease Division, Animal and Plant Quarantine Agency, Gimcheon-si 39660, Republic of Koreassing0805@naver.com (E.-G.S.);; 2Department of Animal Veterinary Development, BioPOA, Hwaseong-si 18469, Republic of Korea

**Keywords:** PRRSV-1, codon pair deoptimization, ORF7, genetic stability, vaccine

## Abstract

Porcine reproductive and respiratory syndrome virus (PRRSV) is major economic problem given its effects on swine health and productivity. Therefore, we evaluated the genetic stability of a codon pair de-optimized (CPD) PRRSV, E38-ORF7 CPD, as well as the master seed passage threshold that elicited an effective immune response in pigs against heterologous virus challenge. The genetic stability and immune response of every 10th passage (out of 40) of E38-ORF7 CPD was analyzed through whole genome sequencing and inoculation in 3-week-old pigs. E38-ORF7 CPD passages were limited to 20 based on the full-length mutation analysis and animal test results. After 20 passages, the virus could not induce antibodies to provide effective immunity and mutations accumulated in the gene, which differed from the CPD gene, presenting a reason for low infectivity. Conclusively, the optimal passage number of E38-ORF7 CPD is 20. As a vaccine, this may help overcome the highly diverse PRRSV infection with substantially enhanced genetic stability.

## 1. Introduction

Porcine reproductive and respiratory syndrome virus (PRRSV) is responsible for reproductive failures in sows and respiratory distress in piglets, ultimately causing considerable economic loss in the pork industry. This disease was introduced to the Republic of Korea in the mid-1980s; since then, it has been a challenging viral disease in the Swine industry due to the emergence of new mutations and vaccine failure [1,2,3].

The RNA virus belongs to the family *Arteriviridae*, genus *Porarterivirus*. The genus is made up of two species: *Betaarterivirus suid 1* (PRRSV-1) and *Betaarterivirus suid 2* (PRRSV-2) [4,5]. The PRRSV-1 contains four genotypes and PRRSV-2 has nine lineages based on ORF5 sequence analysis [6,7]. The PRRSV genome is 15 kb in size, including 11 open reading frames: ORF1a, ORF1a’-TF, ORF1b, ORF2a, ORF2b, ORF3, ORF4, ORF5, ORF5a, ORF6, and ORF7. ORF1a, ORF1a’-TF, and ORF1b can encode non-structural proteins, whereas ORF2-7 can encode for structural proteins (GP2a, E, GP3, GP4, GP5, M, and N proteins). The PRRSV genome is highly variable and has notable genetic and antigenic diversity [8].

Vaccinations with modified live virus vaccines (MLV) are an ongoing control measure used by most swine rearing farms for disease control. These vaccines are synthesized using serial passages in cell culture, where they undergo genetic and antigenic diversity via mutation; however, the chance of reversion to the original virulent form is high as the virus undergoes mutation [9]. There are reports of MLV recombination with circulating wild PRRSV, which generates highly pathogenic strains of PRRSV [10,11,12,13,14,15,16]. Alternatively, killed vaccines can be used; however, they do not elicit a sufficient immune response to completely protect the host [17,18].

A computer-based codon pair de-optimization method (CPD) of attenuation, which incorporates several silent mutations, leads to an inefficient viral gene translation and viral replication, therefore making it safer to use compared to most commercially available MLVs. The concept of CPD vaccines has been applied in several viruses, such as HIV-1, RSV, Poliovirus, and PRRSV [19,20,21,22,23,24,25,26,27,28]. In a previous study, we successfully applied CPD on the NSP1 gene for PRRSV-2 [29,30] and the ORF7 gene of the PRRSV-1 [31]. The PRRSV-1 CPD vaccinated pigs were successfully protected from the heterologous challenge virus CBNU0495 (GenBank No:KY434183).

Infection ability loss due to viral mutation is a common occurrence in vaccine development. As the virus passage progresses in cell cultures, virus attenuation is evident; it is noticeable by a reduction in the virulence of the virus. Hence, it is vital to evaluate the gene mutation site as the number of passages progress in cells to yield improved synthetic RNA virus vaccine candidates [23,32]. Mutations of the virus as the passages progress in vitro have an impact on the immune response in vivo [33]. Thus, identifying the optimal number of passages capable of inducing an effective immune response in vivo during vaccine development is necessary.

Determining an appropriate passage number that will elicit efficient immune responses in vivo without mutation is crucial to determining the master seed virus in the final vaccine composition. Hence, we passaged the E38-ORF7 CPD virus and evaluated mutations (genome sequencing) and immune responses (serum antibody level) by inoculating pigs with a heterologous challenge virus.

## 2. Materials and Methods

### 2.1. Animals

For this study, 30 (*n* = 30) commercially bred three-week-old male pigs were purchased from a PRRSV negative herd. All pigs were negative for PRRSV and porcine circovirus type (PCV) 2, as determined via real-time PCR carried out prior to delivery and arrival. After arrival, the pigs were stratified based on their weight, which was followed by randomized allocation into six groups (*n* = 5 group per group). The animals were housed in separate rooms with an automatic watering system. The animal experiment protocol of this study was approved by QIA Institutional Animal Care and Use Committee (approval number 2021-560); the experiment was conducted as per the guidelines suggested by the Animal Ethics Committee.

### 2.2. Vaccine and Challenge Virus

Codon pair de-optimized PRRSV-1 (E38-ORF7 CPD) was passaged in MARC-145 cells [29,31,34] and cultured in RPMI-1640 medium supplemented with 10% fetal bovine serum (FBS) and 1% penicillin and streptomycin. The SNU 090485 (PRRSV-1; GenBank accession No: JN315686) was isolated from the lungs of aborted porcine fetuses from a farm in Gyeonggi-do, South Korea in 2009 and was used as the challenge virus. The challenge virus was cultured and titrated under the same conditions as PRRSV-1.

The CPD PRRSV-1 virus (E38) and challenge virus (SNU 090485) used in the study belong to Sub-group A under Sub-type 1.

### 2.3. Passages Limits of Attenuated E38-ORF7 CPD

The E38-ORF7 CPD attenuated virus, developed in our lab [31], was serially passaged in MARC-145 cells for up to 40 passages. Virus samples were collected at each 10th passage (after 10, 20, 30, and 40 passages). To determine the maximum limit of the passages effective at protecting animals against the challenge virus, successive passages of the E38-ORF7 CPD vaccine virus were carried out in MARC-145 cells. The full genome sequence of the entire virus was then determined at each 10th passage and was analyzed using the method previously described by Kim et al. [3].

### 2.4. Experimental Design

30 commercially crossbred male three-week-old pigs were randomly divided into 6 groups (*n* = 5 per group. Groups 1 to 4 received the 2 mL 10^5^ TCID50/mL vaccine E38-ORF7 CPD harvested at the 10th, 20th, 30th, and 40th passages, via intramuscular route. Groups 5 and 6 were control groups and did not receive the vaccine. Except for Group 6, all groups were challenged four weeks after vaccination with 3 mL of 10^5^ TCID_50_/mL SNU090484 virus inoculated evenly into both nostrils using a mucosal atomization device. Following the challenge, the animals were observed daily for clinical symptoms, such as fever (rectal temperature), anorexia, inappetence, respiratory distress, nasal secretion, and lameness. The average daily weight gain (ADWG) was calculated based on the body weights recorded on weekly basis from day 0 of vaccination until necropsy. Serum and blood samples were taken for immunological assays (serum IgG ELISA) and viral load estimations were performed on a weekly basis. At the time of autopsy (2 weeks post-challenge or 6 weeks post-vaccination), lung samples were collected for interstitial pneumonitis and antigen distribution observation.

### 2.5. Serology

Serum was collected weekly from all pigs in each group. These samples were tested for the presence of PRRSV specific antibodies using a commercially available PRRSV ELISA kit (HerdCheck PRRS 3XRTM, IDEXX laboratories Inc., Westbrook, ME, USA). The samples were read at an optical density of 650 nm. The antibody cut off value for serum samples—a >0.4 S/P ratio—was considered positive.

### 2.6. Quantification of the Virus in Serum and Lung Tissue

Serum samples were collected 7 and 14 days post-challenge from the various groups to quantify the PRRSV load. The total RNA was extracted and cDNA was synthesized. Reverse transcriptase PCR (RT-PCR) targeting the ORF7 gene was performed using the QuantiTect Probe PCR Kit (Qiagen, Hilden, Germany) to quantify the viral load in the samples. The 5′ATGGCCAGCCAGTCAATCA3′ primer was used as the forward primer and 5′TCGCCCTAATTGAATAGGTGA3′ was used as the reverse primer. After a standard curve was obtained for the primers, amplification was carried out as previously described by Park et al. [35]. The melting curves of the amplified products were analyzed to verify PCR specificity, considering samples with Ct values of <35 as positive samples [36].

### 2.7. Histopathology

At 2 weeks post-challenge, microscopic lesions were analyzed as previously described [37]. For microscopic lesion analysis, samples were fixed in 10% neutral buffered formalin and processed for histological examination. Samples were presented as blinded treatment groups and evaluated by a veterinary pathologist. The interstitial pneumonia lung lesions were scored from 0–3 (0: no lesions; 1: mild interstitial pneumonia; 2: moderate interstitial pneumonia; 3: severe interstitial pneumonia).

### 2.8. Statistical Analysis

The data shown in the manuscript was expressed as the mean ± standard deviation (SD) for triplicate experiments. SPSS (version 16.0) software was used to perform statistical analyses. For animal experimental results, Pearson’s correlation coefficient was used to evaluate the infection and immune paraments. A significance level of *p* < 0.05 was considered for all statistical analyses.

The data collected were assessed via one-way ANOVA followed by Tukey’s multiple comparisons test. A significance level of *p* < 0.05 was considered for all statistical analyses.

## 3. Results

### 3.1. Genetic Variation via Passage Analysis

After transfection of the vaccine strain, successive passages were performed in MARC-145 cells. The full sequence of the E38-ORF7 CPD genome was secured in units of 10 passages. Whole genome analysis revealed a confirmation of the substitution of one of 22 CPD bases at passage 10–40 at the 327th nucleotide position, where nucleotide ‘C’ was substituted for nucleotide ‘T’ (Figure 1). An additional substitution occurred at the 204th nucleotide of the 30th and 40th passage sample, where nucleotide ‘C’ was substituted for ‘T’ (Figure 1). These mutations were silent, which resulted in an unchanged final amino acid (protein). Complete genome sequencing was performed using multiple sequence alignment and genomic characteristics were evaluated. At passage 10, there were seven nucleotide mutations added; a further three were added at passage 20 (accumulated to ten). As passage progressed, the number of mutations accumulated and, finally, there were 19 nucleotide mutations at passage 40. As a result, amino acid substitutions were observed in the ORF2a (94 a.a, Met→Ile) and ORF5 (37 a.a, Asn→Asp) regions at passage 30 and not at the CPD applied region (Table 1).

### 3.2. Evaluation of the Efficacy of the CPD Virus Various Passages

#### 3.2.1. Clinical Evaluation

The rectal temperature of the animals was recorded on a weekly basis. The temperature reading did not exceed 41 °C after the challenge. No significant increase in body temperature was observed in the Control Group. Anorexia, impotence, bleeding, congestion, joint edema, dyspnea, cough, orbital exudate, and severe diarrhea were not observed post-challenge.

#### 3.2.2. PRRSV-Specific Antibodies

Pigs in the Negative Control (NC; Unvaccinated Unchallenged) remained serologically negative throughout the experiment. The passages up to 20 showed 80% seroconversion by the 14th day post-vaccination and 100% seroconversion by the 21st day. Passages 30 and 40 exhibited an antibody positivity rate of 40–60% on the 21st day post-vaccination. The antibody level in the Positive Control (PC; Non-Vaccinated Challenge Group) started to increase from the seventh day post-challenge and increased at 2 weeks post-challenge (Table 2).

#### 3.2.3. Quantification of PRRSV RNA in Serum

Blood samples were collected from the animals in the experimental group on the seventh and 14th days post-challenge. On the seventh day, the post-challenge group, which received the passages 10 and 20, had viral positivity of 20%, whereas the group receiving the passage 30 and 40 had 60% positivity. The PC Group showed 100% virus positivity and the NC Group exhibited 0%. However, the positivity rate of virus was 20% for passages 10 and 20, 60% for passage 30, 100% for passage 40 and the PC, and, finally, 0% for the NC Group.

The Vaccine Group showed considerable differences to the PC (Unvaccinated Challenge) Group until passage 20 on day 14 post-challenge. In Passage 30 and Passage 40 Groups, there was no considerable reduction in the virus number as compared to other PC. The virus was not detected in the animals from the NC Group (Figure 2).

#### 3.2.4. Microscopic Lesions Viral Load in Lung

A collective autopsy was conducted on the experimental animals of the various groups two weeks after the challenge virus inoculation. The degree of interstitial pneumonia and viral load in the lung was assessed. The vaccine group showed a significant difference from the Control Group with respect to interstitial pneumonia and viral load in the lungs only when it progressed until the 20th passage. The PRRSV genomic copy numbers in the PC (unvaccinated challenge) group were 5.46 ± 0.62 (log_10_); in comparison with the PC group, passage 10, 20, and negative had a 0 ± 0 * genomic copy number and there was significant difference from the PC. The viral copy number of Passage 30 Group was 1.66 ± 2.28 * (log_10_), which was significantly reduced compared to the PC group. There were no statistically significant differences between passage 40 (2.94 ± 2.69) (log_10_) and the PC. No viral load was identified in the NC Group (Table 3).

Interstitial pneumonia was evident in the PC Group with a lung lesion score of 1.73 ± 027. The PC Group showed a thickened wall of the lung alveoli with increased mononuclear cell accumulation. The lung sections from the NC were clear and showed no signs of PRRSV pneumonia. Lung lesion scores of passages 10, 20, 30, 40, and the NC were 0.46 ± 0.29 *, 0.53 ± 0.18 *, 1.33 ± 0.33, 1.6 ± 0.27, and 0.13 ± 0.18 *, respectively. There was a statistically significant difference in lung lesions between the passage up to 20 and the NC and PC Groups. Microscopic lung lesions in passages 10 and 20 were nearly identical to the NC Group (Figure 3). The alveolar wall thickening and mononuclear cell accumulation increased as the passage number increased. There was no statistically significant difference between passage 30 and 40 compared to the PC Group (Table 3, Figure 3).

## 4. Discussion

The PRRSV is a major economic disease affecting swine worldwide. The virus undergoes mutations and recombination to generate a diverse range of virus strains [10,13,38]. Biosecurity and vaccinations are currently used under controlled measures. Most swine farms use modified live virus vaccines as opposed to the killed vaccine as they perform better with respect to immune response. However, the modified live virus vaccines have many pitfalls, such as not fully protecting against heterologous strains, reversion to virulent forms, and recombining with the circulating wild virus to generate new variants of the virus [10,11,39].

Designing a modified live virus vaccine with a high safety profile is, therefore, warranted [18]. Codon pair de-optimization generates the MLV vaccine via recoding the viral genome and introducing several synonymous mutations without altering the final protein [19]. The CPD has widely been used to generate diverse attenuated virus vaccines [40]. Recoding viral genomes via synonymous mutations or substitutions provides live attenuated vaccine candidates with low reversion risk [41]. However, their stability under selective pressure is not well explored.

Regardless of the mechanism of attenuation via CPD, the genetic stability of the live attenuated virus vaccine is of paramount importance to avoid the risks of reversion to virulent forms. Thus, it is essential to evaluate the genetic and phenotypic stability through scanning mutations at different levels of passages and evaluating the immune response induced via these various passages in vivo in the experimental animals. Vaccine candidates must be standardized by tracking the genetic stability of the attenuated virus and exposing the virus to strong selective pressure and its immune response in vivo.

Additionally, passage restriction of live vaccine is a requirement for licensure of vaccine and is usually performed in cells; however, the results in vitro cannot be correlated with in vivo immunogenicity in pigs. Hence, we passaged the CPD virus until passage number 40 in MARC-145 cells and every 10th passage was selected for evaluating the immune response in pigs.

Le Nouen et al. [23] evaluated the genetic stability of genome scale de-optimized RNA virus vaccine candidates under selective pressure via serial passages at progressively increasing temperatures. The CPD virus is attenuated and temperature sensitive. They reported that a mutant with higher mutations was not deattenuated and remained stable, whereas a CPD virus with a low number of mutations lost temperature sensitivity and attenuation.

In this study, mutations in the virus were evident as the passages progressed in the MARC-145 cell culture system. Every 10th passage was selected to check the genetic stability by sequencing the virus genome, with immune responses evaluated in experimental animals. In general, there is a difference in the virus proliferation in MARC-145 cells in vitro and experimental animals.

Passage 10, in vitro passage virus exhibited single amino acid change in the ORF1a, ORF1b, ORF2a, ORF3 and ORF5, whereas Passage 20 (in vitro) showed amino acid variation at ORF1a and ORF4. Although there was change in the amino acid in the region other than the CPD site, the immune response generated through passages 10 and 20 is significantly high compared to passages 30 and 40.

At virus passage 30, the whole genome sequence of the virus at various passages revealed there were substitutions in the ORF2a and ORF5 gene, which is a different site to that of the target (ORF7) gene. Although it is not certain that the infectivity decreased due to ORF5 (37aa, Asn→Asp) change, previous reports indicated that the ORF2a region (94aa, Met→Ile) is known to be a substitution that occurs when PRRSV-1 strain adapts to MARC-145 cells [42]. The ORF2a gene encodes for GP2, which is a minor glycosylated structural protein essential for virus infectivity. This mutation may alter the structural confirmation of the GP2 protein. Consequently, this may result in reduced virus replication, leading to slow viral growth in vivo and reduced immune response. This change in the virus is evident in a previous study performed on chickens, in which the passages progressed in cell culture [43]. The study also revealed the gradual change in the virus population during serial passage in chickens and chick embryo fibroblast cells. The gradual change in the virus population was due to the change in the amino acid residue at the position 253 of VP2 protein.

In general, attenuated viruses are characterized by slow growth in vivo with antibody seroconversion that occurs between 14 and 25 days post-vaccination. Here, the pigs that received attenuated viruses of passages 30 and 40 showed only 40–60% seroconversion by day 28 of inoculation. This indicates viral proliferation difficulty in the host. The viral load reduction post-challenge was reduced in both groups receiving viruses of passage 30 and 40. Reduced serum IgG was evident in the group vaccinated with passage 30 and 40 as compared to passages 10 and 20 of the CPD vaccine. Reduced serum IgG levels were also reflected by the viral copy number and lung lesions score.

The reduction in attenuation of the virus may also be due to the cumulative mutation introduced in the other genes of the virus. Previous reports also indicated the role of mutations in other regions of the virus in the diattenuation of CPD viruses [23]. However, a single mutation in a different gene is sufficient to change the virus. Infectivity loss is a common occurrence during vaccine development [33]. Hence, it is important to establish the appropriate passage conditions during the vaccine master seed production. We found that our CPD vaccine is stable for up to 20 passages with maintenance of the effective immunity.

## 5. Conclusions

Codon pair de-optimized viral vaccines are generated via recoding the gene of the infectious organism to incorporate several silent or synonymous mutations. The genetic stability and in vivo immune response of such modified genome is not known. In our study, we have successfully attenuated the PRRSV-1 (Subtype 1A) virus by applying CPD on ORF7 gene. The genetic stability of the developed vaccine candidate was assessed after successful passage through the MARC-145 cells via whole genome sequencing of the random passages. The immune responses of such random passages in the vaccinated pigs were evaluated after challenges with the virulent virus. The maximum passage number to demonstrate the genetic stability without change in the effective immune response was passage number 20.

## Figures and Tables

**Figure 1 viruses-15-01071-f001:**
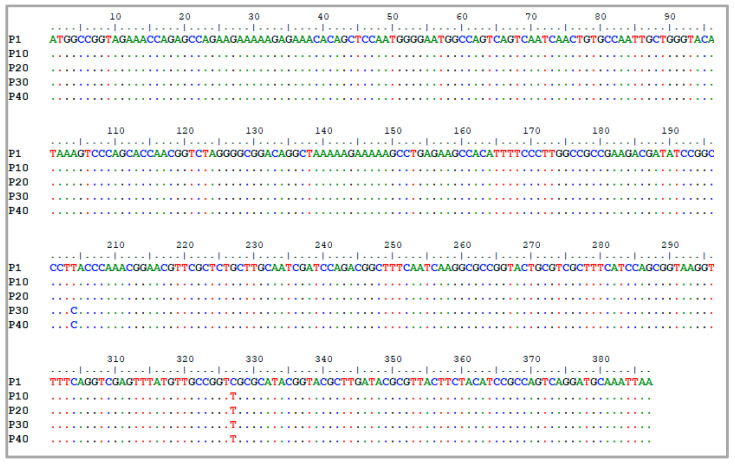
Sequence alignment serially passaged E38-ORF7 CPD in MARC-145 cells. The samples were accordingly denoted E38-ORF7 CPD P10, E38-ORF7 CPD P20, E38-ORF7 CPD P30, and E38-ORF7 CPD P40. At every 10th passage, viral genome was screened for mutation.

**Figure 2 viruses-15-01071-f002:**
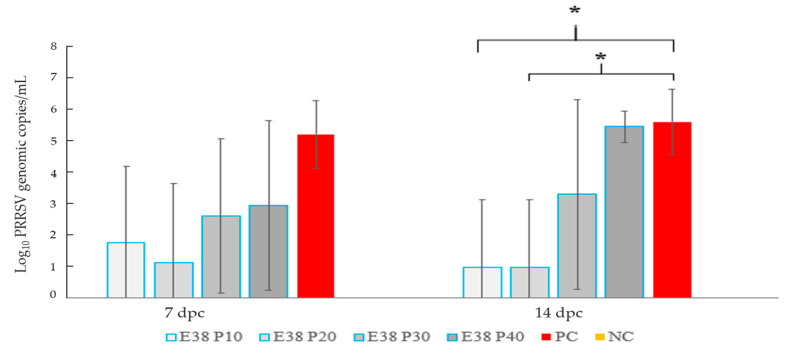
Bar graph depicting viral load quantification of serum samples from pigs. A Ct value of <35 cycles was considered positive. No significant difference was observed between different groups on seventh day post-challenge. On 14th day post-challenge, there were statistically significant differences between the 10th and 20th passage compared to the rest of the groups. (*) indicates significant differences where *p <* 0.05. NC: negative control; PC: positive control; dpc: day post challenge.

**Figure 3 viruses-15-01071-f003:**
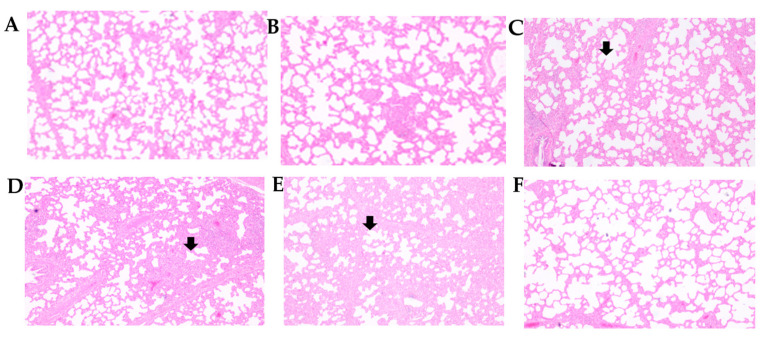
Histopathological lesions of necropsied pig lungs stained with Hematoxylin and Eosin at 14 dpc immunized at different passage numbers of E38-ORF7 CPD vaccine. (**A**) Passage 10 (P10), (**B**) Passage 20 (P20), (**C**) Passage 30 (P30), (**D**) Passage 40 (P40), (**E**) Positive control (PC, unvaccinated challenge group), (**F**) Negative Control (NC, Unvaccinated Unchallenged Group). Severe alveolar thickening and mononuclear infiltration was noticed in the Passage 30, Passage 40 and Positive Control Groups (indicated by an arrow mark). Passage 10 and 20 lung lesions were almost similar to Negative Control Group.

**Table 1 viruses-15-01071-t001:** Complete genome sequence analysis of serially passed E38-ORF7 CPD virus *^a^*.

	Passage 10	Passage 20	Passage 30	Passage 40
**Number of nt change**	7	10	16	19
nucleotide	ORF1a (2941nt, C->T) ORF1a (6778nt, G->A) ORF1b (4090nt, T->C) ORF2a (509nt, T->C) ORF3 (214nt, A->G) ORF5 (50nt, T->C) ORF7 (327nt, C->T)	ORF1a(2613nt, C->A) ORF1a(4990nt, A->G) ORF4(529nt, T->G)	ORF1a (5814nt, A->G) ORF2a (282nt, G->A) ORF2a (591nt, G->A) ORF4 (519nt, C->T) ORF5 (109nt, A->G) ORF7 (204nt, T->C)	ORF1a (721nt, C->T) ORF1a (2981nt, A->G) ORF1a (3624nt, T->C)
**Number of a.a change**	6	+3 (9)	+2 (11)	+2 (13)
a.a	ORF1a (981a.a, Arg->Cys) ORF1a (2226a.a, Asp->Asn) ORF1b (1364a.a, Tyr->His) ORF2a (170a.a, Val->Ala) ORF3 (72a.a, Ser->Gly) ORF5 (17a.a, Phe->Ser)	ORF1a (871a.a, Ser->Arg) ORF1a (1664a.a, Thr->Ala) ORF4 (177a.a, Phe->Val)	ORF2a (94a.a, Met->Ile) ORF5 (37a.a, Asn->Asp)	ORF1a (241a.a, Leu->Phe) ORF1a (994a.a, Gln->Arg)

*^a^* According to the serial passage in MARC-145 cells, gradual nucleotide and a.a change was observed. At passage 30, the full genome of virus was sequenced and the result revealed that there were substitutions in the ORF2a and ORF5 genes.

**Table 2 viruses-15-01071-t002:** PRRSV-specific antibody at different passages and in PC and NC *^a^*.

	0 dpi	7 dpi	14 dpi	21 dpi	28 dpi	35 dpi	42 dpi
Passage 10	0 0%	0 0%	0.46 80%	0.68 100%	0.73 100%	1.07 100%	1.15 100%
Passage 20	0 0%	0 0%	0.41 80%	0.59 100%	0.71 100%	0.91 100%	1.16 100%
Passage 30	0 0%	0 0%	0 0%	0.3 60%	0.31 60%	0.58 80%	1.48 100%
Passage 40	0 0%	0 0%	0 0%	0.23 40%	0.28 40%	0.44 80%	1.24 100%
PC	0 0%	0 0%	0 0%	0 0%	0 0%	0.08 20%	1.14 100%
NC	0 0%	0 0%	0 0%	0 0%	0 0%	0 0%	0 0%

*^a^* Results are displayed as percentage of positive pigs.

**Table 3 viruses-15-01071-t003:** Lung lesion score and tissue PRRSV genomic copies from different groups.

	Passage 10	Passage 20	Passage 30	Passage 40	PC	NC
Lung lesion score	0.46 ± 0.29 *	0.53 ± 0.18 *	1.33 ± 0.33	1.6 ± 0.27	1.73 ± 0.27	0.1 3± 0.18 *
Log_10_ PRRSV genomic copies/g	0 ± 0 *	0 ± 0 *	1.66 ± 2.28 *	2.94 ± 2.69	5.46 ± 0.62	0 ± 0 *

(*) indicates significant differences compared to the PC (Unvaccinated Challenge) Group. Interstitial pneumonia in the lung lesions were scored from 0–3 (0: no lesions; 1: mild interstitial pneumonia; 2: moderate interstitial pneumonia; or 3: severe interstitial pneumonia). The PRRSV genomic copy number was estimated through reverse transcription PCR (RT-PCR) targeting the ORF7 gene. The viral copy number was expressed as log PRRSV genomic number/g of tissue. PRRSV was defined as porcine reproductive and respiratory syndrome virus, while NC was defined as negative control; PC was defined as positive control.

## Data Availability

The datasets generated or analyzed during this study are available from the corresponding author on reasonable request.

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
