# Peer review of "Evaluation and Determination of a Suitable Passage Number of Codon Pair Deoptimized PRRSV-1 Vaccine Candidate in Pigs"

_viruses, 2023, doi:10.3390/v15051071_

Round 1

Reviewer 1 Report

1.The author directly uses PRRSV from cells as vaccine, didn't use any adjuvants to treat with the virus?that‘s is very imprecise.

2.The author's results after the virus attack are obviously not enough, and the amount of data is not enough to support his conclusion.

3. Author descriped that the 20th generation has an immune protection effect.  The first generation have an immune protection effect?  if the first generation has some protection effect, all of the work should be reconsider.

Author Response

Dear Editor:

We thank you for your time and consideration on our submission (viruses-2258628). Below we address the editorial office’ comments and list of changes that we made to our manuscript according to their comments. The original referee comments are provided in black color, whereas our answers are given in blue.

Reviewer 2 Report

1. In the materials and methods section 2.3, 3-week-old piglets: (1) provide details about the selection criteria for selected pigs and commercial farm. (2) which was the mean BW of selected animals? (3) More details about the breed and vaccination program of the commercial farm. (4) Were vaccinated the selected piglets?

2. In the results section 3.2.2, it should be tested levels of PRRSV neutralizing antibodies in serum.

3. In the results section 3.2.3, in addition to detection of viral RNA in serum, it is also required to detect viral RNA in lymph node tissue and alveolar macrophages. PRRSV is mainly distributed in these tissues or cells of pig.

4. In the results section 3.2.3, it should be added the pathological picture of Lung lesion in the text.

5. In the discussion section, “Designing a modified live……not well explored.”, you should add appropriate references in the text.

6. In the discussion section, “Le Nouen et al. (2017)……”, “……by Ya-maguchi et al. (2000).”, the reference number should be placed in parentheses.

Author Response

(The authors gave the same response as above.)

Reviewer 3 Report

This manuscript describes the use of an experimental vaccine strain of PRRSV-1 in pigs, comparing different passages of the strain for efficacy.

The authors try to relate the differences in protection afforded by each preparation to changes in nucleotide and amino acid sequences, but this is not clearly demonstrated. Further discussion and a revision of the conclusions is therefore needed.

Specific comments:

Introduction (or materials and methods section 2.1) - The subtypes of the viruses used should be described.

Materials and methods, section 2.1 - Please list the source of the MARC-145 cells.

Materials and methods, section 2.1 - The method for generating the CPD virus must be described, since it has not been referenced elsewhere.

Materials and methods, section 2.2 - Please list the sex(es) and breed of the pigs used.

Materials and methods, section 2.2 - Please describe in more detail the method for intranasal infection (what devices were used, was just one nostril used or both, etc.).

Materials and methods, section 2.2 - Please provide details of how the lung samples were obtained and selected.

Materials and methods, section 2.3 - This appears to duplicate the information in section 2.2?

Materials and methods, section 2.5 - Please provide more detail on how the standard curve was produced.

Discussion, paragraph 5 - This isn't fully clear - were sequence differences found after proliferation in vivo, or were differences in growth kinetics seen in vitro?

Discussion, paragraph 6 - The nucleotide change seen in ORF7 at passage 10 is not discussed here. There should be more discussion about the relative importance of the changes seen at each passage in this study.

Discussion and Conclusion - Overall it is not clear what contribution the sequence changes after passage 20 make to the observed changes in the in vivo infections using passage 30 or passage 40 material.

Conclusion - The last sentence indicates that there are no mutations at passage 20, which is not true.

Author Response

(The authors gave the same response as above.)
